# Dietary Exposure to 2,2′,4,4′-Tetrabromodiphenyl Ether (BDE-47) Causes Inflammation in the Liver of Common Carp (*Cyprinus carpio*) and Affects Lipid Metabolism by Interfering with Steroid Hormone Biosynthesis Pathways

**DOI:** 10.3390/ijms262010152

**Published:** 2025-10-18

**Authors:** Shuhuang Chen, Nian Han, Yujie Huang, Huimin Sun, Youlian Liu, Defang Chen, Zhiqiong Li, Xin Zhang

**Affiliations:** Department of Aquaculture, College of Animal Science and Technology, Sichuan Agricultural University, Chengdu 611130, China; pipyakas@126.com (S.C.); 18303629527@163.com (N.H.); hyjscnydx@163.com (Y.H.); 17653477846@163.com (H.S.); lyl20210225@163.com (Y.L.); chendf_sicau@126.com (D.C.); lizhiqiong454@163.com (Z.L.)

**Keywords:** PBDEs, transcriptomics and metabolomics, multi-omics integrated analysis, steroid hormone metabolism, inflammatory response

## Abstract

2,2′,4,4′-tetrabromodiphenyl ether (BDE-47) is a common environmental contaminant and widely detected in aquatic surroundings, while only a few reports exist on the hazard mechanism in economic aquatic animals. It has been shown that 40 and 4000 ng/g of BDE-47 dietary exposure over 42 days significantly increased the levels of blood triglycerides, glucose, and liver glycogen in carp (*Cyprinus carpio*). Tissue observations showed that BDE-47 resulted in vacuolation, atrophy, and fat deposition in hepatocytes. Combined metabolomic and transcriptomic analyses revealed that BDE-47 affected the inflammatory response and the biosynthesis of steroid hormones. This was further confirmed by gene expression related to inflammatory factors (*il-10*, *tnf-α*, *il-1β,* and *tgf-β1*), lipid metabolism (*acc*, *fas*, *srebp*, *rxr*, *atgl*, *hsl*, and *lpl*), and the steroid hormone biosynthetic pathway (*11bhsd*, *hsd3b*, and *star*). Thus, BDE-47 affects liver inflammatory response and lipid deposition through steroid hormone biosynthesis in carp. This helps us to understand how BDE-47 dietary exposure impacts inflammation and lipid metabolism in fish, which affects the health of aquaculture and has potential risks to human health.

## 1. Introduction

For decades, polybrominated diphenyl ethers (PBDEs) have been extensively employed as flame retardants [1] and although most have been withdrawn from the market, they persist in the environment due to bioaccumulation and high toxicity [2]. PBDEs can exist in the natural environment for several months to several years [3]. The most prevalent and toxic PBDE is 2,2′,4,4′-tetrabromodiphenyl ether (BDE-47) [4,5]. The recorded content of BDE-47 in the Yangtze River Basin is 42.9 ng/g dw [6], while significantly greater concentrations of BDE-47 3800 ng/g dw have been detected in sediment samples collected near electronic waste processing facilities [7]. Studies have been conducted on the noxiousness of BDE-47 to the liver, resulting in changes such as hypertrophy [8], cytoplasmic vacuolation, and apoptosis of hepatocytes [9,10]; however, relevant physiological metabolic data and the toxicological mechanisms remain unknown.

The liver is central to regulating glucose and lipid metabolism in animals [11]. Existing investigations have revealed that BDE-47 could disrupt lipid and glucose metabolism in animals. Dietary intake of BDE-47 induces severe hepatic lipid degeneration in mice [12], while in vitro experiments have demonstrated that treatment with BDE-47 affects cellular lipid metabolism in a hepatocellular carcinoma cell line (HepG cells) [13]. Mice exposed to BDE-47 exhibit a two-fold increase in liver and blood triglyceride levels [14]. Moreover, BDE-47 prevents glucose homeostasis by diminishing glucose absorption in the liver, pancreas, and adipose tissue, concurrently reducing production and secretion of insulin by beta cells, and inhibiting the expression of glucose transporters [15]. Additionally, BDE-47 affects the differentiation of 3T3-L1 adipocytes via peroxisome proliferator-activated receptor gamma (PPARγ), leading to disrupted glucose and lipid metabolism [14,16]. In contrast to mammals, fish rely on a diverse array of energy sources, such as fats, carbohydrates, and proteins, which may have different metabolic effects in response to BDE-47 [17,18]. A study of zebrafish reported that BDE-47 simulates estrogens through its influence on the expression of related genes within liver cells [19], which is different from what has been observed in mammals. However, few studies have examined how BDE-47 affects carbohydrate and lipid metabolism in fish.

Food intake is considered one of the main routes through which vertebrates are exposed to PBDEs [20]. BDE-47 has a high n-octanol-water distribution and is enriched in the sediment [21,22,23,24]; thus, fish that live below the middle layer of the aquatic surroundings are more threatened [25]. Common carp (*Cyprinis carpio*) is a widely distributed fish in Asia and worldwide [26,27]. It lives in the middle- and lower-water environments [28] and usually ingests organic debris from the bottom mud [26,29]; thus, carp are vulnerable to the toxicity of BDE-47. In common carp, pancreatic tissue is embedded within the hepatic parenchyma. In our previous study, after 28 days of exposure, the liver of the carp began to show partial lesions, and by 42 days, the lesions were very obvious, so we chose the duration of 42 days [30]. To investigate the effect of BDE-47 on lipid metabolism in carp, we measured biochemical markers in the blood and liver of carp with exposure to environmentally relevant concentrations of BDE-47, performed histopathological observations, and performed transcriptomic and metabolomic analyses to identify key pathways and enzymes. Gene expression at different time points was examined to confirm how BDE-47 affects lipid metabolism. This study provides preliminary insights into the effect of BDE-47 on the metabolism of carp triglycerides, and offers a certain theoretical basis and scientific evidence for environmental protection and fish health.

## 2. Results and Discussion

### 2.1. BDE-47 Exposure Results in the Accumulation of Lipids in the Carp Liver

BDE-47 is a commonly used flame retardant that easily accumulates in the sediment of lakes and rivers. A previous report showed that high levels of BDE-47 (3800 ng/g dw) [7] were detected in sediment from rivers near e-waste treatment plants in Vietnam, and pollutant levels increased between 2012 and 2014. The PBDE content in lake sediments is 52.1 ng/g dw in the middle reaches of the Yangtze River, in which the heaviest proportion is 42.9 ng/g·dw BDE-47 [6]. BDE-47 has been confirmed to be neurotoxic and have endocrine-disrupting effects in animals, and has been extensively studied in mammals [31], birds [32], and fish [33]. In addition, mammalian studies have shown that BDE-47 significantly affects glucose and lipid metabolism [34,35], but its effects on other animals have not been determined. Carp live below the middle level of the water, ingest organic matter from the bottom mud or sediment [36,37,38], and are thus susceptible to BDE-47. However, there are few reports on the effects of BDE-47 in carp. Thus, to explore the effect of BDE-47 on glycolipid metabolism in carp, in this study, we fed carp environmentally relevant concentrations of BDE-47 for 42 days.

This study used a DMSO group as a solvent-control group. No significant changes in plasma glucose (Figure 1A, *p* = 0.625), triglycerides (Figure 1B, *p* = 0.160), or total cholesterol (Figure 1C, *p* = 0.349) were observed between the control group and the DMSO group after the carp were fed BDE-47 for 42 days. Feeding the high BDE-47 dose (4000 ng/g) caused a significant increase in plasma glucose (Figure 1A), triglycerides (Figure 1B), and liver glycogen (Figure 1D). Similarly, feeding the low BDE-47 dose (40 ng/g) revealed a significant increase in plasma glucose and liver glycogen levels (Figure 1A,D). These results are consistent with previous findings that BDE-47 affects blood and liver triglyceride homeostasis in mice [12,39]. In addition, BDE-47 induces a hepatic nonalcoholic fatty liver phenotype in rats. Despite these changes associated with glucose and lipid metabolism, BDE-47 did not significantly affect cholesterol content in the carp liver. In contrast, epidemiological research has reported a positive correlation between blood PBDE concentrations and cholesterol levels, which was also reported in the seal (*Phoca largha*) [40]. However, similar to the findings of this study, no significant change in the cholesterol level was detected in European starlings (*Sturnus vulgaris*) after BDE-47 exposure [41]. This study reports on fish for the first time, but it remains unclear whether this difference is due to species differences.

In addition, the results from the hematoxylin–eosin staining show that normal tissue structure can be observed in the liver cells of the DMSO and control groups, with the liver cells arranged closely together, being nearly round in shape and having clear cell boundaries, with nuclei located centrally and being either round or oval (Figure 2A). This indicates that DMSO in the experimental feed did not affect the liver structure of the carp. After exposure to BDE-47, the distribution of liver cells became irregular, cell boundaries became blurred, and vacuoles appeared in liver cells, with more severe vacuole formation in the high-concentration group (Figure 2A). The results indicate that both low and high concentrations of BDE-47 cause damage to cellular structure, with higher concentrations leading to more severe damage. The structural damage further impacts cellular function. Oil Red O staining revealed that the liver cell morphology in the control and DMSO groups was normal, with nuclei appearing blue. Compared to the control group, the low-concentration group exhibited a small amount of red lipid droplets in the cytoplasm of carp liver cells, while the red lipid droplets in the cytoplasm of carp liver cells treated with high concentrations significantly enlarged, and lipid droplet deposition worsened, indicating that the degree of fatty degeneration further increased (Figure 2B), consistent with the H&E staining results. In addition, we used Image-Pro Plus 6.0 software to analyze the optical density of red lipid droplets in the livers of the groups (n = 9, the number of samples per group is 9.) to evaluate fat deposition (Figure 2C). The results indicated that no significant difference was observed between the control group and the DMSO group, but a significant increase was detected in lipid deposition in the liver of carp in the low- and high-concentration groups (Figure 2C, *p* = 0.041). We also validated the correlation between the results of Oil Red O and the biochemical results. We found that the internal clustering among different treatment groups was relatively stable, while there were significant separation differences between different treatment groups. The number of lipid droplets had a strong correlation with plasma glucose, triglycerides, total cholesterol, and liver glycogen (Appendix A). The results in Figure 1 indicate abnormal cell metabolism. Combining Figure 2A,B, we can infer that BDE-47 exposure leads to abnormal lipid metabolism function in cells. Similar observations of tissue damage have been reported in other animals. For example, BDE-47 causes mitochondrial damage in mouse hepatocytes and worsens hepatic steatosis in mice with nonalcoholic fatty liver [42,43]. Furthermore, in vitro experiments have demonstrated that BDE-47 disrupts the normal morphology and function of numerous liver cells, such as HepG2 cells, human embryonic stem cells differentiated into functional hepatocellular-like cells, and Human Hepatocyte Progenitor Cells (HepaRG cells) [43,44]. However, in fish, only zebrafish [43] and crucian carp [45] have been reported to be susceptible to BDE-47, which causes histological lesions in the liver, preventing a rapid recovery. Taken together, BDE-47 caused structural abnormalities and lipid deposition in the liver of carp, leading to disorders of glucose and lipid metabolism.

### 2.2. BDE-47 Affects the Metabolites Involved in Glucose and Lipid Metabolism in the Carp Liver

The DMSO treatment had no significant effect on glucose or lipid metabolism in carp. Thus, in subsequent metabolomic and transcriptomic experiments, we used the DMSO group as a control group to explore the crucial enzymes and pathways contributing to the liver lipid deposition caused by BDE-47. The metabolome consists of intermediates and end metabolites that provide direct insight into the response of an organism to stress. Therefore, we used untargeted metabolomics to assess the chronic effects of BDE-47 on carp after exposure to an environmentally-relevant density of BDE-47, and key metabolites were identified through data mining. OPLS-DA was used to determine the metabolic differences between the BDE-47-treated and DMSO groups (Figure 3A,B). A PLS-DA permutation test (n = 200) was conducted to evaluate the validity of the OPLS-DA model (Figure 3C,D). These results collectively demonstrated the reliability of the OPLS-DA model in both negative and positive ion modes. Then, we compared the liquid chromatography/mass spectrometry data with the HMDB online database and determined that lipids and lipid molecules accounted for the highest proportion of metabolites among the three groups, comprising 179 types (34.29%) (Figure 3E). Subsequently, pathway topology analysis, impact score assessment, pathway enrichment analysis (Student’s *t*-test), *p*-value, and pathway physiological function analyses revealed that the differentially abundant metabolites in the low-concentration group were primarily involved in ascorbate and lysine degradation, aldarate metabolism, primary bile acid biosynthesis, and other pathways (Figure 3F), while these metabolites were mainly involved in riboflavin metabolism, tryptophan metabolism, leucine, isoleucine, and valine biosynthesis and other pathways in the high-concentration group (Figure 3G). Studies on aquatic animals are limited in this area, as we performed a metabolomic investigation of BDE-47 treatment in fish for the first time. Consistent with our results, metabolome analyses revealed that BDE-47 affects glucose and lipid metabolism in earthworms, leading to an increase in lactic acid accumulation [46,47,48]. In addition, BDE-47 inhibits the TCA cycle and aerobic respiration in mussels [49,50]. In experiments with more advanced animals, BDE-47 caused dysfunction of lipid metabolism in the white adipose tissue of mice, possibly by inhibiting *β*-oxidation, fatty acid uptake, and resynthesis [47,51]. These results provide new evidence regarding the metabolomic effects of BDE-47 in aquatic animals and serve as an important reference for further exploration of its impact on glucose and lipid metabolism in fish.

### 2.3. Transcriptome Data Reveal That BDE-47 Affects Lipid Accumulation Through Glucose- and Lipid Metabolic-Related Pathways

To explore the proposed mechanism of the effect of BDE-47 on glucose and lipid metabolism in the carp liver, transcriptomic analysis was conducted after 42 days of being exposed to BDE-47 via diet (Figure 4A–F). The correlation coefficients between biological replicates in the different groups were > 0.89, indicating similarity in the components within the groups and confirming the reliability of the differentially expressed genes (DEGs) (Figure 4A,B). In addition, GO and KEGG enrichment analysis of DEGs was performed (Figure 4C–F). Compared with the DMSO group, the DEGs in the low-concentration group were mainly enriched in GO items, such as nicotinamide phosphoribosyltransferase activity, the chitosan degradation process, chitinase activity, the chitin metabolism process, and immune system process regulation (Figure 4C). The DEGs between the high-concentration group and the DMSO group were mainly enriched in GO items such as the glucosamine degradation process, the glucosamine degradation process, the chitin degradation process, chitinase activity, and the chitin metabolism process (Figure 4D). The results of the KEGG enrichment analysis revealed that pathways related to glucose and lipid metabolism (fatty acid biosynthesis, fatty acid degradation, and glycolysis/gluconeogenesis), and amino acid metabolism were more enriched in the low-concentration sample than in the DMSO sample (Figure 4E). The DEGs in the high-concentration sample were enriched in lipid metabolism (cholesterol metabolism, absorption, and fat digestion, the PPAR signaling pathway, steroid hormone biosynthesis, and pyruvate metabolism) and amino acid metabolism pathways compared to those in the DMSO group (Figure 4F).

Similarly, Ahmed Khalil reported that BDE-47 affects liver triglyceride metabolism in mice [52]. Transcriptome analysis of HepG2 cells revealed that BDE-47 significantly affects glycolysis/gluconeogenesis. Another study conducted by Joshua F Robinson on human placental cytotrophoblasts reported that the DEGs affected by BDE-47 were related mainly to lipid and steroid metabolism, including pathways involved in estrogen and aldosterone [53]. In fish, only one such case was reported in zebrafish. Transcriptome analysis showed that treating zebrafish embryos with BDE-47 alters metabolic pathways, such as fatty acid metabolism, purine metabolism, glycolysis, and gluconeogenesis. The zebrafish study showed enrichment in the steroid hormone biosynthetic pathway, aligning with the present study. Steroid hormones, particularly glucocorticoids, are closely linked to glucose and lipid metabolism. Glucocorticoids promote glucose production, inhibit glucose utilization, increase lipid breakdown, and stimulate fatty acid release, while also promoting fatty acid synthesis and storage, contributing to fat accumulation. Thus, steroid hormone biosynthesis may regulate lipid metabolism after BDE-47 exposure [54,55]. PPAR act as intracellular lipid sensors that regulate lipid metabolism under various physiological and pathological conditions. When nutrients are abundant, PPARγ regulates adipocyte differentiation and lipid storage; during energy deficiency, PPARα promotes fatty acid β-oxidation and enhances lipolysis by activating the transcription factor TFEB. In response to lipid overload and stress, PPARβ/δ collaborates with PPARα to inhibit lipid synthesis and restore lipid autophagy [56].Taken together, the results of this study showed that BDE-47 potentially affects lipid accumulation through pathways associated with glucose and lipid metabolism, including the steroid biosynthetic pathway, the PPAR signaling pathway, fat digestion and absorption, and so on.

### 2.4. BDE-47 Induces an Inflammatory Response in the Liver of Carp

Metabolomics: KEGG enrichment showed that tryptophan metabolism and glycerophospholipid metabolism were changed in the low-concentration group (Figure 3F). Tryptophan metabolism directly or indirectly affects inflammation through three pathways [57]. The level of glycerophospholipid metabolism can reflect whether the inflammatory response in the body is unbalanced [58]. Purine metabolism was significantly changed in the high-concentration group (Figure 3G). Purines are able to modulate inflammatory mechanisms [59]. Transcriptomics: GO enrichment of the detected DEGs showed significant changes in the regulation of immune system process, immune response, and so on in the low-concentration group (Figure 4C). When the body’s immune system is destroyed, it induces an inflammatory response. The organic acid metabolic process and heme binding in the high-concentration group were significantly changed (Figure 4D). Organic acids are capable of damaging the intestinal mucosa and inducing an inflammatory response [60]. Activation of TLR4 signaling, induced by the direct physical interaction between heme and the MD2/TLR4 complex, results in the exacerbation of inflammation [61]. KEGG enrichment of the detected DEGs showed significant changes in the necroptosis and JAK-STAT signaling pathway in the low-concentration group (Figure 4E). Studies in mice have shown that liver senescence is interrelated with enhanced necroptosis that leads to chronic inflammation of the liver [62]. The JAK-STAT pathway is a signaling pathway that regulates gene expression associated with inflammation, immunity, and cancer [63]. Significant changes in the PPAR signaling pathway and ferroptosis were found in the high-concentration group (Figure 4F). PPAR-α [64] and PPAR-β/δ [65] and PPAR-γ [66] can affect the inflammatory response. Ferroptosis can be used as a marker in response to inflammation [67]. Based on these results, we suggest that BDE-47 induces an inflammatory response in carp, so we quantified the gene expression of related inflammatory factors.

As can be seen from the Figure 4, in the control sample and the DMSO group, there was no significant change in gene expression levels, indicating that DMSO does not significantly affect gene expression. After 3 days of exposure, expression levels of pro-inflammatory factors *tnf-α*, *tgf-β1*, and *il-1β* in the low- and high-concentration groups significantly increased (Figure 5A–C). The expression level of the anti-inflammatory factor *il-10* significantly decreased (Figure 5D). Moreover, the differences became more pronounced with an increase in exposure time. Previous studies in our lab have shown that exposure to BDE-47 significantly upregulated the expression of *tnf-α*, *tgf-β1*, and *il-1β* [68], which aligns with our experimental results. Analogous results have been found in other animals. Currently, studies on the effect of BDE-47 on animal inflammatory responses are mainly focused on mice. Studies in mice have shown that BDE-47 treatment results in an upregulation of *il-1β* expression in the liver [69]. Similar studies have been conducted on BDE-209, which leads to a significant upregulation of *tnf-α* expression in the intestines and liver of mice. Research in the hippocampus has shown that exposure to BDE-209 significantly decreases the gene expression of *il-10,* whereas *tnf-α* and *il-1β* significantly increased [70], paralleling the results of this study. In summary, our results demonstrate that treatment with BDE-47 alters inflammation-related gene expression in the liver of carp, with more pronounced differences as exposure time increases, suggesting that BDE-47 exposure may result in an imbalance in inflammatory responses within the body.

### 2.5. Transcriptome and Metabolomic Joint Analyses Reveal That the BDE-47 Pathway Induces Lipid Deposition in the Carp Liver

This research investigated the mechanism of the response of the carp liver to BDE-47 exposure by integrating transcriptomic and metabolomic analyses. Correlation analysis revealed strong correlations between many genes and metabolites in the DMSO group compared to those in the low- and high-concentration groups. KEGG pathway co-enrichment analysis demonstrated that 19 and 26 common pathways were enriched, respectively (Figure 6A,B). The co-enriched pathways in the low-glucose and DMSO groups were primarily related to lipid metabolism, including steroid hormone biosynthesis, arachidonic acid metabolism, bile acid biosynthesis, and glycerolipid metabolism (Figure 6A). Pathways associated with glycolysis/gluconeogenesis (TCA cycle), the pentose phosphate pathway, fatty acid degradation, steroid hormone biosynthesis, and oxidative phosphorylation were involved in the high-concentration and DMSO groups (Figure 6B). Similarly, transcriptomic and proteomic studies of HepG2 cells have shown that lipid metabolism and signaling markers, such as the PI3K/AKT/mTOR pathway, significantly affect metabolism, leading to lipid accumulation [71,72]. Moreover, BDE-47 induces significant changes in liver lipid metabolic pathways and inhibits β-oxidation of fatty acids in mice with breast cancer. Thus, the multi-omics results indicate that BDE-47 affects energy metabolism and support the possibility that BDE-47 affects lipid metabolism in fish.

In addition, this study revealed enrichment of the steroid hormone biosynthetic pathway after BDE-47 exposure (Figure 6C). We further analyzed the metabolites and genes related to the steroid hormone biosynthetic pathway, including progesterone, cortisol, and corticosterone, and related genes, such as *hsd3b*, which were significantly upregulated. The only study on zebrafish reported similar results to those of this study; the 21-day 500 µg/L BDE-47 treatment significantly inhibited signaling factors related to steroid hormone synthesis and secretion by the ovaries and brains of zebrafish [71]. In vitro experiments have revealed that BDE-47 affects signal transduction in the steroid hormone pathway of HAC15 cells [73]. Furthermore, BDE-47 increases aldosterone and cortisol levels in HAC15 cells and upregulates the expression of related synthetic genes (*hsd3b2, cyp11b1*, *cyp11b2*, and *mc2r*) [74]. Similarly, the expression of genes associated with steroid hormone metabolic pathways was significantly upregulated in human hepatocytes after exposure to 1 or 10 μm BDE-47 for 48 h. Therefore, the present study further confirmed that BDE-47 induces lipid accumulation in fish by regulating the steroid hormone biosynthetic pathway.

### 2.6. BDE-47 Induces Lipid Deposition Through the Steroid Hormone Biosynthetic Pathway in the Carp Liver

To further determine whether BDE-47 induces lipid deposition through the steroid hormone biosynthetic pathway in the liver of fish, the present study examined the expression levels of related genes in the liver at different time points during the 42 days, including genes related to fat synthesis (*srebp*, *rxr*, *fas*, and *acc*), lipolysis (*atgl*, *hsl*, and *lpl*), steroid hormone biosynthesis (*star*, *hsd3b*, and *hsd3b11*), and related transcription factors (*ppar*, *pi3k*, and *akt*) (Figure 7A and Appendix A). The results showed that *srebp* increased significantly on day 3 (*p* < 0.017), whereas the fat synthesis factor levels did not change on days 1 and 3 after the BDE-47 treatment (Figure 7A and Appendix A). However, *fas* and *star* were significantly upregulated nearly 40-fold on days 7 and 14. The expression of these genes decreased on days 28 and 42 but compared to the control group, it remained higher (Figure 7A and Appendix A). Similarly, *Tigriopus japonicus* exposed to BDE-47 exhibited significantly increased expression of *acc* and *srebp* [75], and an experiment suggested that the hepatotoxicity of BDE-47 in L02 cells may be mediated by RXR [76]. Prior studies have indicated that BDE-209 induces hepatic lipid deposition by upregulating *fas*, *acc*, *rxr*, and *glut1* in rodents, suggesting partial similarity among PBDE-induced lipid deposition [77].

Although *atgl* and *hsl* were upregulated on day 42, no significant differences were found in lipolysis genes. Previous studies confirm that *lpl*, *atgl*, and *hsl* are key in lipolysis, with *lpl* and *atgl* converting triglycerides to diglycerols, and *hsl* releasing fatty acids [78].

Together, these factors regulate triglyceride breakdown. No studies have explored lipolysis factor expression in animals exposed to BDE-47, suggesting that BDE-47 may not cause lipid deposition through triglyceride breakdown. Steroid hormone biosynthesis factor expression increased significantly on days 7 and 14 (*p* < 0.003). Variations in steroid hormone metabolic factor expression exist across studies, with BDE-47 notably increasing *17β-hsd* and *star* transcription levels in human follicular cells [77], while BDE-47 significantly inhibits the expression of *cyp11a1*, *pparγ*, *star,* and *3β-hsd* in mouse ovarian cells [79]. In addition, BDE-47 has no effect on *3β-hsd* in human placental chorionic cancer cells. These discrepancies may have arisen from various factors, such as differences in research subjects, experimental conditions or sample sources. In fish, similar to the findings of the present study, BDE-47 induces the upregulation of *cyp19a* in zebrafish ovaries [80]. Taken together, these findings indicate that BDE-47 promotes the transcription of genes related to steroid hormone metabolism in the carp liver. Furthermore, a correlation analysis was conducted on the expression levels of these genes, and a strong correlation was detected between the steroid hormone biosynthesis pathway factors and fat synthesis genes (Figure 7B). Therefore, these results further support the notion that BDE-47 promotes fat synthesis by affecting the steroid hormone biosynthetic pathway leading to lipid deposition in the liver of carp.

## 3. Materials and Methods

### 3.1. Chemical and Experimental Fish

In the Introduction, we mentioned that PBDEs can exist in the natural environment for several months to years [3]. Additionally, BDE-47 can accumulate in biological organisms [2]. To observe the chronic toxic effects of BDE-47 in carp, we chose to conduct a long-term experiment. In our previous research, we found that after 28 days of exposure, carp livers began to show some lesions, and by 42 days, the lesions became very obvious, which is why we chose 42 days as our observation period [30].

Previous studies have indicated that feeding is an important route for animals to intake BDE-47 [20]. In aquatic environments, BDE-47 tends to accumulate more in sediment [21,22,23,24]. Carp are fish that live in the mid to lower water layers, so they come into contact with more sediment [25]. Therefore, the low and high concentrations of BDE-47 we used were referenced from concentrations found in water environment sediment and in areas with serious environmental pollution. The concentration of BDE-47 in the Yangtze River Basin is 42.9 ng/g dry weight [6], while in more polluted areas, such as near electronic waste disposal facilities, the concentration is 3800 ng/g dry weight [7].

In summary, the method of exposure to BDE-47 used in this experiment was through diet. BDE-47 is poorly soluble in water, we dissolved it in DMSO before mixing it with the diet. The exposure duration was 42 days, with exposure concentrations of 40 and 4000 ng/g.

The analytical standard BDE-47 (HPLC ≥ 98% purity) was bought from Yuanye Biotechnology Co., Ltd. (Shanghai, China). Then, 400 mg of BDE-47 was dissolved in 50 mL DMSO and mixed with 150 mL water to produce a BDE-47 solution of 2 mg/mL. The BDE-47 solution was mixed with standard commercial feed (Rongchuan Feed Co., Ltd., Zigong, China) and air-dried to produce final concentrations of 0, 40, and 4000 ng/g.

The carp were purchased from a carp farm in Chengdu, Sichuan Province, and temporarily kept in disinfected fish tanks at a depth of 0.5 m. We chose healthy carp that swam normally and had no diseases. The weight of the carp we used was 22.10 ± 1.30 g and the season was autumn. The growth and developmental characteristics of carp at this stage are quite significant. The water temperature was maintained at 20 °C, and the water was oxygenated to maintain dissolved oxygen concentrations above 6.0 mg/L, and these fish were fed regularly at 14:00. A 12 h light and 12 h dark cycle was followed every day. After a two-week acclimation period, formal experimentation commenced. All of the experimental procedures were approved by the Animal Health and Use Committee of the Chinese Academy of Sciences and Sichuan Agricultural University (SAAS20212020 and DKY-S2021202072). The approval date is 20 October 2022.

### 3.2. Experimental Design

The experiment was divided into 4 groups: blank control group, DMSO group (solvent control group), 40 ng/g BDE-47 group (low-concentration group), and 4000 ng/g BDE-47 group (high-concentration group), with 3 sets in each group, totaling 20 carp per sets, amounting to 360 fish. The carp were randomly assigned to aquariums, with 30 fish in each tank. During the experiment, continuous oxygenation and natural light were maintained, and the carp were fed 3% of their body weight in feed at 14:00 daily. Half an hour after feeding, any uneaten feed and feces were removed. The liver and blood of the carp were collected on day 42 post-feeding, with 12 fish collected from each set. After collection, samples were quenched in liquid nitrogen for more than 15 min and then frozen at −80 °C for subsequent physiological and biochemical index determination, tissue section preparation, and omics detection.

Additionally, to investigate the effect of the feeding duration of BDE-47 on carp, this study also divided healthy carp into 4 groups, with 3 sets in each group, and 10 fish per set. The livers of carp were collected on days 1, 3, 7, 14, 28, and 42 post-feeding, with 3 fish collected from each group for subsequent qPCR detection.

### 3.3. Biochemical Analysis

This study used the Jiancheng reagent kits (A154-1-1, A110-1-1, A111-1-1, A043-1-1, Jiancheng Bio, Nanjing, China) from Nanjing to determine the glucose, triglyceride, and total cholesterol levels in carp plasma, as well as the glycogen content in liver tissue.

### 3.4. Histopathological Analysis

H&E

The livers randomly selected from four carp were removed and fixed with 4% paraformaldehyde for 24 h. Subsequently, the organs were washed and dehydrated with different concentrations of ethanol. Subsequently, these livers were hyalicized in xylene and encapsulated in paraffin. Paraffin blocks were cut into 5 μm slices using a German Leica RM2125 rotary microtome (Leica, Nussloch, Germany). After dewaxing, the slices were stained with hematoxylin and eosin. Finally, the slices were observed and photographed using an ECLIPSE 200 microscope from Nikon in Japan (Nikon, Tokyo, Japan), and the images were analyzed with Image-Pro Plus 6.0. The magnification was 20×. Three biological replicates were performed in each treatment group, and the number of three tissue sections was analyzed for each biological replicate, for a total of 36 sections, and a total of 6 histological regions were examined.

Oil Red O

After the liver were excised, they were fixed in 10% neutral formalin solution for 10 min, washed 2–3 times in PBS, and the surface moisture was blotted dry with absorbent paper. The liver from the same location were placed on a fixation tray and quickly frozen in a cryostat. Once the tissue was sufficiently frozen, the liver were sliced into thin sections using the cryostat, placed on a glass slide for 10 min, and then stained. The sections were immersed in modified Oil Red O staining solution for 15–20 min, then washed with isopropanol and distilled water, followed by hematoxylin staining for 5 min, and finally differentiated and rinsed. After staining, the slides were mounted with glycerin gelatin for observation and photography. The magnification was 40×. The concentration of IDO in the lipid droplets was measured using Image-Pro Plus 6.0.

### 3.5. Validation of the Correlation Between Oil Red O and Biochemical Data

Multidimensional quantitative data (quantification of Oil Red lipid droplets and detection values of multiple biochemical indicators) were collected from different treatment groups. Dimensionality reduction was performed using the Principal Component Analysis (PCA) algorithm to draw PCA score plots, visually presenting the clustering patterns and separation degree among different groups. SPSS 22.0 statistical software (SPSS, Inc., Chicago, IL, USA) was used to perform correlation analysis of the quantitative data of Oil Red lipid droplets and biochemical indicators based on the Pearson method, and heatmaps were visualized using the method at https://www.chiplot.online/ (accessed on 25 September 2025).

### 3.6. Metabolomic Analysis

For each concentration, 3 sets were used, and for each set, the livers of 6 fish were used for the experiment, with each liver weighing 50 mg. The samples were ultrasonically extracted and then centrifuged. The mixture was dried for 3UPLC-MS analysis, which was performed using Progenesis QI software (v3.1) (WatersACorporation, Milford, MA, USA). Positive and negative data were combined and analyzed on the 3Majorbio I-Sanger Cloud Platform (www.i-sanger.com (accessed on 25 September 2025)). Partial least squares discriminant analysis (PLS-DA) was used to identify significant changes in metabolites between the control group and the treatment group, with an error detection rate (FDR) < 0.01 and variable impact projection (VIP) >1. In addition, there were significant changes in metabolites between groups (FDR < 0.01, VIP > 1), which are called DMs. Venn diagrams and heatmaps were generated to visualize changes in metabolites between different groups.

### 3.7. Transcriptomic Analysis

We performed transcriptomic sequencing of carp liver in three groups exposed on 42 days, and the detailed steps are referenced [68]. Total RNA was extracted using an animal tissue total RNA extraction and purification kit (B518651, Sangon Biotech, Shanghai, China) following the manufacturer’s protocol. The concentration of total RNA was assessed using a Bioanalyzer (Agilent 2100, Santa Clara, CA, USA) and 1% agarose gel electrophoresis, and only RNA with RIN ≥ 7.0, OD260/280 ≥ 1.8, and OD260/230 ≥ 1.5 was selected for subsequent steps. Library construction and sequencing were conducted by Shanghai Majorbio Bio-pharm Technology Co., Ltd. (Shanghai, China). To ensure the reliability of the sequencing results, SeqPrep (https://github.com/jstjohn/SeqPrep (accessed on 25 September 2025)) and Sickle (https://github.com/najoshi/sickle (accessed on 25 September 2025)) were utilized to filter the raw reads and remove low-quality sequences (those sequences longer than 5 nucleotides). After obtaining clean reads, de novo assembly was performed using Trinity (https://github.com/trinityrnaseq/trinityrnaseq/wiki (accessed on 25 September 2025)) to generate the longest non-redundant Unigene set, which was then compared against six databases for annotation information. The six databases used were the following: NR (https://www.ncbi.nlm.nih.gov/refseq/about/nonredundantproteins/ (accessed on 25 September 2025)), Swiss-Prot (https://www.ExPASy.org/resources/swiss-model (accessed on 25 September 2025)), Pfam (http://pfam.xfam.org/ (accessed on 25 September 2025)), COG (https://www.ncbi.nlm.nih.gov/research/cog-project/ (accessed on 25 September 2025)), GO (http://www.org (accessed on 25 September 2025)), and KEGG (http://www.genome.jp/kegg/ (accessed on 25 September 2025)). Subsequently, software from http://deweylab.github.io/RSEM/ (accessed on 25 September 2025) was used to compare and estimate the expression abundance of the assembled Unigenes. After normalizing the read counts between different samples, differential expression genes (DEGs) were filtered using DEGseq software (version 1.60.0) based on *p* adjust < 0.001 and |log2FC| ≥ 1. To further analyze the biological functions of the DEGs, GO and KEGG pathway functions were annotated and enriched for the significantly up-regulated and down-regulated genes (*p* < 0.05).

### 3.8. Integration Analysis of the Transcriptome and Metabolome

Correlation analysis of genes and metabolites according to the Pearson method based on transcriptome and metabolome data. log2FC| > 1 is considered a differentially expressed metabolite. *p* < 0.01 was considered to indicate a significant correlation. Nine-quadrant plots were plotted to analyze the correlation between transcripts and metabolites. Based on KEGG enrichment analysis of the transcriptome and metabolome, the co-enrichment pathway was identified. Pathways related to lipid metabolism were selected for pathway heatmap mapping and KEGG pathway mapping. Subsequently, the FPKM values and normalized peak intensity values of the metabolites in the pathway were converted with z-scores, and the heat map was mapped using TBtools (version 2.325).

### 3.9. qPCR

Q-PCR was performed on the related genes of carp liver at different time points. For detailed steps, please refer to Zhang et al. [81]. The results were normalized to that of *β-actin* and *gadph*, which were used as the internal controls. The gene expressions were calculated by the 2^−∆∆CT^ method. The primers used are shown in Appendix A.

### 3.10. Statistical Analysis

The data processing involved in this study refers to the methods used in Li et al. [68]. The data in this study are presented as mean ± SEM. Statistical analysis was performed using SPSS 22.0 software (SPSS, Inc., Chicago, IL, USA). One-way ANOVA was used to calculate inter-group differences, and Duncan’s method was used for multiple comparisons. *p* < 0.05 was considered statistically significant.

### 3.11. Distribution of Experiment Personnel

The group allocation work was managed by the head of the experimental design; the conduct of the experiment was the responsibility of the experiment personnel; and the outcome assessment was handled by those involved in data evaluation, ensuring that all data were correctly categorized and analyzed according to groups. The data analysis was conducted by data analysts who performed further statistical analysis based on the grouping situation to ensure the reliability and scientific validity of the experimental results. We ensured that appropriate personnel supervised and verified the grouping situation at all stages to avoid any bias or error, and that all processes strictly adhered to the experimental protocol and standards.

## 4. Conclusions

In conclusion, this in vivo study showed that BDE-47 exposure leads to inflammation and lipid accumulation in the carp liver. Metabolomic and transcriptomic analyses revealed changes in metabolites related to glycerophospholipid metabolism, steroid hormone biosynthesis, and inflammatory response. Multi-omics analysis showed that BDE-47 affects lipid synthesis-related genes through the steroid hormone biosynthetic pathway. The study highlights distinct lipid deposition mechanisms in fish and mammals, offering insights into BDE-47′s effects on fish metabolism and environmental protection.

However, it should be noted that the concentration of BDE-47 in the feed cannot be directly equated to the actual concentration in the carp, as the accumulation of pollutants may vary significantly among different individuals compared to the concentration in the feed. Therefore, in the design of similar experiments in the future, it is recommended to include measurements of the actual concentration of BDE-47 in the relevant target tissues to enhance the scientific rigor of the experimental design and the validity of the conclusions.

## Figures and Tables

**Figure 1 ijms-26-10152-f001:**
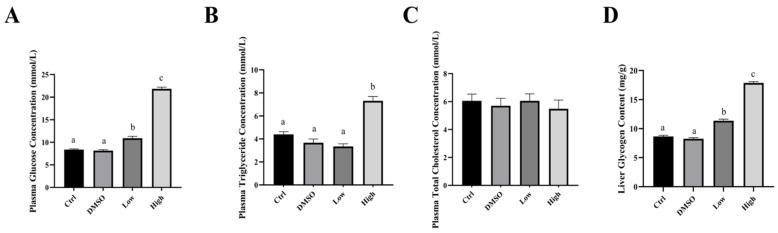
The biochemical parameter changes in the blood and liver and histological changes in the liver of carps in the control group, DMSO group, low-concentration group (40 ng/g BDE-47), and high-concentration group (4000 ng/g BDE-47) after feeding them the BDE-47 diet for 42 days. The plasma glucose content (**A**), triglyceride content (**B**), total cholesterol content (**C**), and hepatic glycogen content (**D**) of the carp liver. All data are presented as mean ± standard error of measurement (SEM), and different letters (a–c) indicate significant differences. The letters from a to c represent significance from low to high, and if different groups have the same letter, it means that there is no significant difference between the two groups.

**Figure 2 ijms-26-10152-f002:**
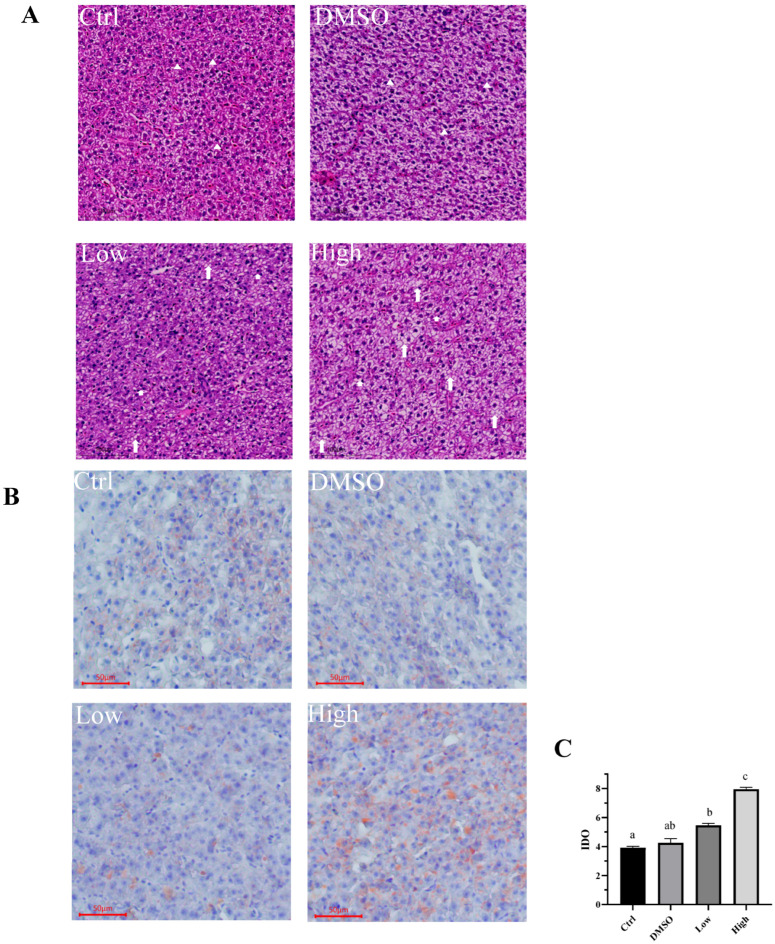
In H&E staining (**A**), the white triangles represent normally shaped 165 cells, the long arrows indicate vacuolated cells, and the short arrows represent cells with unclear boundaries. In oil Red O staining (**B**), the lipid droplets of oil red O staining were quantified by Image Pro Plus 6.0 (n = 9, (**C**)). All data are presented as mean ± standard error of measurement (SEM), and different letters (a–c) indicate significant differences. The letters from a to c represent significance from low to high, and if different groups have the same letter, it means that there is no significant difference between the two groups.

**Figure 3 ijms-26-10152-f003:**
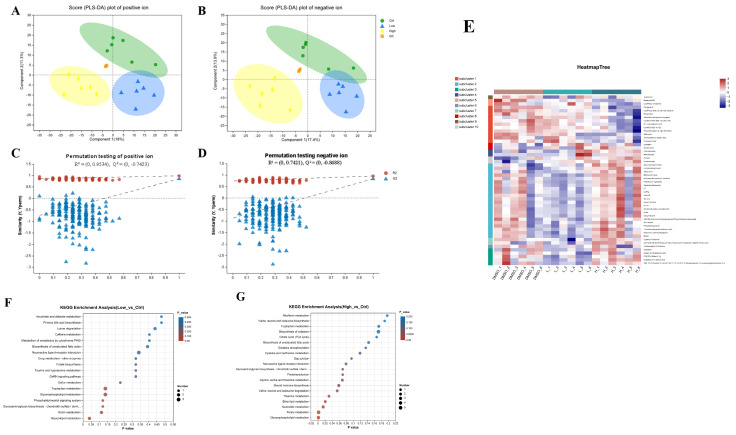
Confidence analysis of metabolome. OPLS-DA model (**A**,**B**) and PLS-DA substitution test (**C**,**D**) for positive ion cations (**A**,**C**) and negative ion (**B**,**D**). Heat map and cluster analysis of differential metabolite content (**E**). The names of metabolites are listed on the right, the clusters of metabolites on the left, and the clusters of samples on the top. L stands for the low-concentration group (40 ng/g), H stands for the high-cncentration group (4000 ng/og). The metabolome (**F**,**G**) changes of carp liver after 42 days of BDE-47 dietary exposure. KEGG enrichment of differential metabolites observed in the low-concentration group (**F**) and high-concentration group (**G**) compared with the DMSO group.

**Figure 4 ijms-26-10152-f004:**
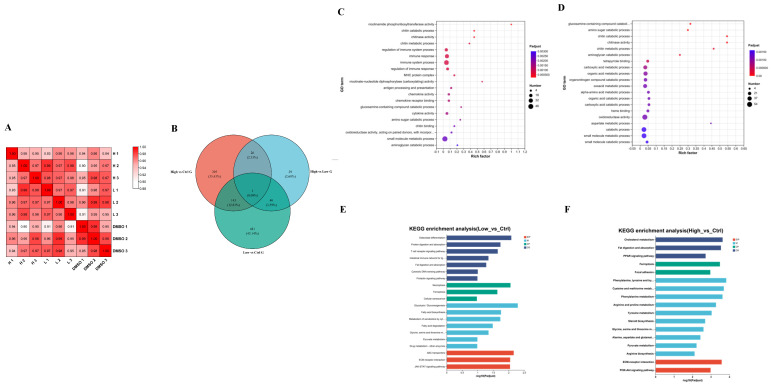
The size of the bubble represents the amount of metabolites enriched in the pathway, and the color of the bubble represents the *p* value. Correlation analysis of sample transcriptomes and Venn diagram of DEG amounts (**A**,**B**). H indicates 4000 ng/kg BDE-47 dietary exposure group, L indicates 40 ng/kg BDE-47 dietary exposure group, and DMSO indicates DMSO control group. The transcriptome changes of carp liver after 42 days of BDE-47 dietary exposure. GO enrichment of DEGs observed in the low-concentration group (**C**) and high-concentration group (**D**) compared with the DMSO group, and KEGG enrichment of DEGs observed in the low-concentration group (**E**) and high-concentration group (**F**) compared with the DMSO group.

**Figure 5 ijms-26-10152-f005:**
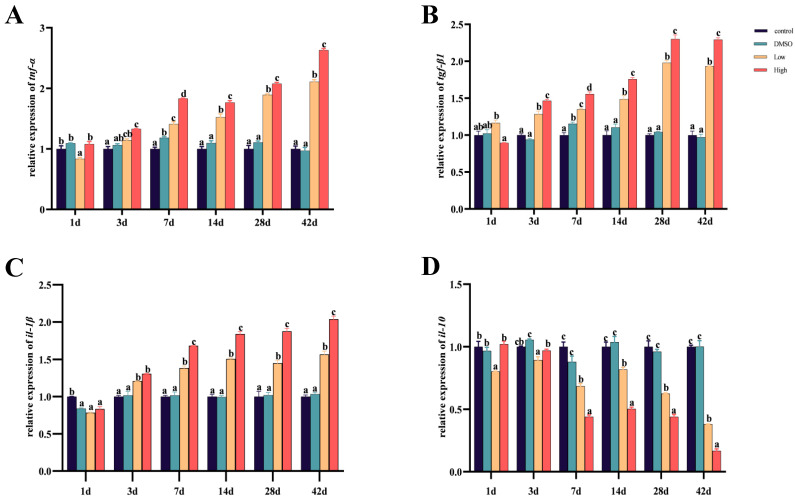
After 1, 3, 7, 14, 28, and 42 days of BDE-47 dietary exposure, the expression levels of inflammation-related genes *tnf-α* (**A**), *tgf-β1* (**B**), *il-1β* (**C**), and *il-10* (**D**) in the liver of the carp were detected. In the figure, black represents the control group, green represents the DMSO group, yellow represents the low-concentration group (40 ng/g BDE-47), and red represents the high-concentration group (4000 ng/g BDE-47). The x-axis shows different time points, while the y-axis represents the relative expression levels of the genes. All data are presented as mean ± standard error of measurement (SEM), with different letters (a–d) indicating significant differences. The letters from a to d indicate significance from low to high; if different groups have the same letter, it signifies no significant difference between the two groups.

**Figure 6 ijms-26-10152-f006:**
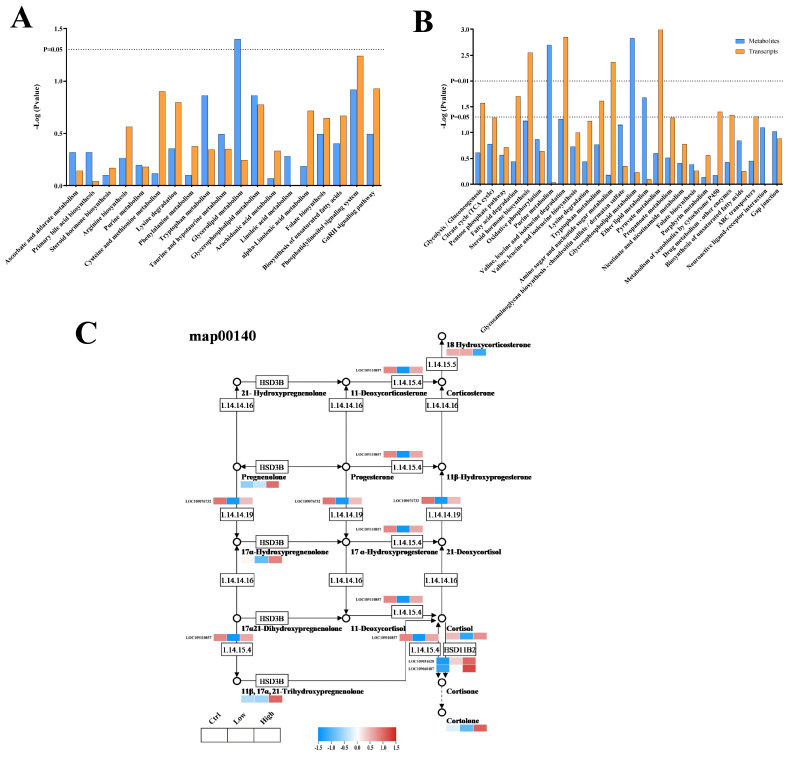
Integrated analysis of metabolome and transcriptome. (**A**,**B**) All of the KEGG pathways co-enriched in the low- (**A**) and high-concentration (**B**) groups. Blue columns represent the KEGG pathways enriched in the metabolome, orange columns represent the KEGG pathways enriched in the transcriptome, and the vertical axis represents the -Log2 *p* value for each pathway. (**C**) Detailed analysis of the genes and metabolites in the steroid hormone biosynthesis pathway (map00140), which was enriched in both the low-concentration group and the high-concentration group. The first cell of all cells represented the DMSO group, the second cell represented the low-concentration group, and the third cell represented the high-concentration group. The color from red to blue indicated the level of expression. All of the genes are presented with their Gene IDs, and all of the metabolites are annotated with their compound names.

**Figure 7 ijms-26-10152-f007:**
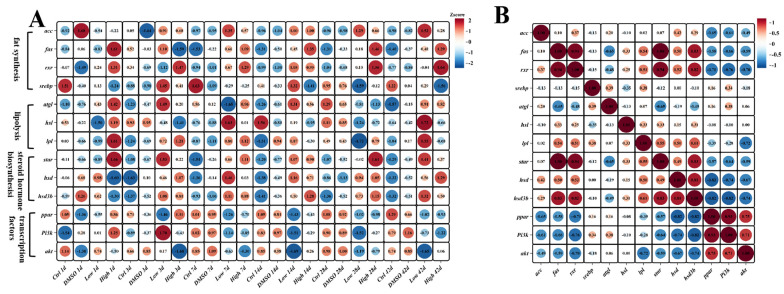
After 1, 3, 7, 14, 28, and 42 days of BDE-47 dietary exposure, the expression levels of lipid metabolism-related genes in the liver of the carp were detected (**A**), and the correlation between the expressions was analyzed (**B**). (**A**) qPCR detection of the expression levels of key genes in lipid synthesis, decomposition, and steroid hormone biosynthesis pathways in carp liver. A z-score was used to normalize the gene expression levels, and the color, from red to blue, indicates the gene expression levels from high to low. (**B**) A Pearson correlation coefficient was used to analyze the correlation of gene expression levels. The color, from red to blue, indicates the correlation of gene expression levels from high to low.

## Data Availability

Data is contained within the article or Appendix A.

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
