# Peer review of "Dietary Exposure to 2,2′,4,4′-Tetrabromodiphenyl Ether (BDE-47) Causes Inflammation in the Liver of Common Carp (Cyprinus carpio) and Affects Lipid Metabolism by Interfering with Steroid Hormone Biosynthesis Pathways"

_ijms, 2025, doi:10.3390/ijms262010152_

Round 1
Reviewer 1 Report
Comments and Suggestions for Authors
In this manuscript, the authors explored the mechanisms related to BDE-47-induced inflammation and lipid deposition in carp hepatopancreas. The manuscript is conform to the scope of this journal, but there are still some issues.
Major Issues:
- The significance of the experiment stated in the introduction is not suitable, suggest to adjust.
- Why use "dietary exposure" instead of "environmental exposure"?
- Please describe the detailed process of preparing the solution in 2.1.
- What are the detailed analysis methods in 2.5 and 2.6?
Minor Issues:
- In Figure 3D, there are excessive letters.
- In 3.3, please introduce the role of the involved pathways.
- Appear for the first time in the manuscript, please provide their full names, such as line 56 "HepG cells".
- Line 146, the" ( "should be deleted.
- Line 407, the"17β-hsd" and "star" should be italicized.
- Line 510, the "van" should be "Van".
Therefore, this manuscript is suitable for published to the International Journal of Molecular Sciences after minor repair.
Author Response
We have uploaded the revised manuscript in the attachment, including the version with tracked changes and the version without tracked changes.
|
Response to Reviewer 1 Comments
|
||
|
1. Summary |
|
|
|
We sincerely thank the reviewer and the editorial team for their time and effort in reviewing this manuscript. We are deeply grateful for the constructive comments and profound suggestions provided by everyone. We have made comprehensive revisions accordingly.We have carefully studied all the review comments and responded to each point. Thank you once again for your help! |
||
|
2. Point-by-point response to Comments and Suggestions for Authors |
||
|
Comments 1: The significance of the experiment stated in the introduction is not suitable, suggest to adjust. |
||
|
Response 1: Thank you for your suggestion. We have made modifications to line 77-79, “This study provides preliminary insights into the effect of BDE-47 on the metabolism of carp triglycerides, and offers a certain theoretical basis and scientific evidence for environmental protection and fish health.” Thank you once again for your valuable suggestion. |
||
|
Comments 2: Why use "dietary exposure" instead of "environmental exposure"? |
||
|
Response 2: Thank you for your question. We have added the reference information for “the reason use dietary exposure in Line 63-64”. In this reference, dietary intake is considered one of the main routes through which vertebrates are exposed to PBDEs. We sincerely appreciate your question once again. Linares V, Bellés M, Domingo J L. Human exposure to PBDE and critical evaluation of health hazards [J]. Archives of Toxicology, 2015, 89(3): 335-356. |
||
|
Comments 3: Please describe the detailed process of preparing the solution in 2.1. |
||
|
Response 3: Thanks for your reminding. We have supplemented the detailed amounts of DMSO and water used in line 83-84.“400 mg of BDE-47 was dissolved in 50 ml DMSO and mixed with 150 ml water to produce a BDE-47 solution of 2 mg/mL.”Thank you again for your reminder. |
||
|
Comments 4: What are the detailed analysis methods in 2.5 and 2.6? |
||
|
Response 4: Thanks for your question. We have added the detailed process of the experiment in 2.6 and 2.7. “2.6. Metabolomic Analysis For each concentration, take 3 parallels, and for each parallel, use the hepatopancreas of 6 fish for the experiment, with each hepatopancreas weighing 50 mg. The samples were ultrasonically extracted and then centrifuged. The mixture was dried for 3UPLC-MS analysis, which were analyzed using Progenesis QI software (WatersACorporation, USA). Positive and negative data were combined and analyzed on the3Majorbio I-Sanger Cloud Platform(www.i-sanger.com).Partial least squares discriminant analysis (PLS-DA)was used to identify significant changes inmetabolites between the control group and the treatment group, with an errordetection rate (FDR) <0.01 and variable impact projection (VIP) >1. In addition, there were significant changes in metabolites between groups (FDR < 0.01, VIP > 1), which are called DMs. Venn diagrams and heatmaps were generated to visualize changes inmetabolites between different groups. 2.7. Transcriptomic Analysis We performed transcriptomic sequencing of carp hepatopancreas in three groups exposed on 42 days, and the detailed steps are referenced[32]. Total RNA was extracted using an animal tissue total RNA extraction and purification kit (B518651, Sangon Biotech, China) following the manufacturer's protocol. The concentration of total RNA was assessed using a Bioanalyzer (Agilent 2100, USA) and 1% agarose gel electrophoresis, and only RNA with RIN ≥ 7.0, OD260/280 ≥ 1.8, and OD260/230 ≥ 1.5 was selected for subsequent steps. Library construction and sequencing were conducted by Shanghai Majorbio Bio-pharm Technology Co., Ltd. To ensure the reliability of the sequencing results, SeqPrep (https://github.com/jstjohn/SeqPrep) and Sickle (https://github.com/najoshi/sickle) were utilized to filter the raw reads and remove low-quality sequences (those sequences longer than 5 nucleotides). After obtaining clean reads, de novo assembly was performed using Trinity (https://github.com/trinityrnaseq/trinityrnaseq/wiki) to generate the longest non-redundant Unigene set, which was then compared against six databases for annotation information. The six databases used were: NR (https://www.ncbi.nlm.nih.gov/refseq/about/nonredundantproteins/), Swiss-Prot (https://www.ExPASy.org/resources/swiss-model), Pfam (http://pfam.xfam.org/), COG (https://www.ncbi.nlm.nih.gov/research/cog-project/), GO (http://www.org), and KEGG (http://www.genome.jp/kegg/). Subsequently, software from http://deweylab.github.io/RSEM/ was used to compare and estimate the expression abundance of the assembled Unigenes. After normalizing the read counts between different samples, differential expression genes (DEGs) were filtered using DEGseq software based on P adjust < 0.001 and |log2FC| ≥ 1. To further analyze the biological functions of the DEGs, GO and KEGG pathway functions were annotated and enriched for the significantly up-regulation and down-regulation genes (P < 0.05).” We appreciate your valuable question. |
||
|
Comments 5: In Figure 3D, there are excessive letters. |
||
|
Response 5: Thanks for your reminding. We have corrected figure 3D and submitted the corrected figure in “figures”.We have corrected the figure in line 313.We sincerely appreciate your reminder. |
||
|
Comments 6: In 3.3, please introduce the role of the involved pathways. |
||
|
Response 6: Thanks for your suggestion. We have added the relevant content and reference in line 387-398. We truly appreciate your suggestion once more. |
||
|
Comments 7: Appear for the first time in the manuscript, please provide their full names, such as line 56 "HepG cells". |
||
|
Response 7: Thanks for your comment.“HepG cells” full name is “Hepatocellular carcinoma cell line,we have added it in line 49. Similar problems also occurred in line 287 “HepaRG cells”, its full name is “Human Hepatocyte Progenitor Cells”.We have corrected it. We sincerely appreciate your helpful feedback. |
||
|
Comments 8: Line 197, the" ( "should be deleted. |
||
|
Response 8: Thank you for your reminding. We have deleted it in line 207. We appreciate your reminding. |
||
|
Comments 9: Line 407, the"17β-hsd" and "star" should be italicized. |
||
|
Response 9: Thanks for your careful reminding. We have corrected it in line 541-542. We really appreciate your reminding. |
||
|
Comments 10: Line 510, the "van" should be "Van". |
||
|
Response 10: Many thanks for your suggestion. We have corrected it in line 627. Thank you again for your suggestion. |
||

Reviewer 2 Report
Comments and Suggestions for Authors
The manuscript lacks sufficient evidence that the authors are fully conversant with the biological and pathological characteristics of the chosen fish model. Relevant details regarding the experimental fish (e.g., biometric data, age, sex ratio, and season of collection) are not reported, and the experimental design is not adequately described in the Materials and Methods section. Furthermore, the authors rely on a secondary citation for the histopathological scoring method employed to assess liver pathology across experimental groups, without providing sufficient details of the scoring system or criteria used. These details are crucial for reproducibility and critical interpretation of the results. It should also be emphasised that although routine histopathology on paraffin-embedded sections and light microscopy represents the gold standard in diagnostic histopathology, it constitutes only an entry-level approach in toxicologic pathology research. Ideally, semithin resin-embedded sections observed under light microscopy and, optimally, ultrathin sections evaluated with transmission electron microscopy (TEM) should be adopted to elucidate the pathophysiological mechanisms underlying the reported lesions. Moreover, the literature contains numerous reports describing qualitative and quantitative methods applied to both paraffin-embedded and semithin resin-embedded sections of carp, with advanced image analysis techniques (e.g., fractal analysis and texture analysis) employed to objectively assess liver pathology in toxicological pathology contexts. The authors are invited to refer to them. In addition, the histopathological figures presented in the manuscript are of insufficient dimension, magnification, and resolution to allow proper evaluation of the lesions described. Higher-quality images, preferably including scale bars and representative fields at appropriate magnification, are required to substantiate the authors’ claims. Moreover, there is no evidence of correlation between the morphological observations and the biochemical or metabolomic data. Consequently, the study currently appears as a collection of separate methodological approaches rather than an integrated morpho-functional investigation. The inclusion of exploratory multivariate analyses (e.g., PCA, PLS-DA) linking morphological, biochemical, and omics data could greatly enhance the interpretative strength of the study and provide a more coherent picture of the toxicological impact of BDE-47 in carp.

Author Response
We have uploaded the revised manuscript in the attachment, including the version with tracked changes and the version without tracked changes.
|
Response to Reviewer 2 Comments
|
||||||||||||
|
1. Summary |
|
|
||||||||||
|
Thank you for taking the valuable time and effort on this manuscript, especially for your valuable comments on the organizational pathology, which made the description of the experimental methods in our manuscript clearer. We have carefully read your comments and provide the following responses: 1. We have supplemented the detailed information regarding the experimental fish, and provided a detailed response in Comments 3. 2. We have rewritten the experimental design in the manuscript, with a detailed response in Comments 4. 3. Regarding the issues of histopathology, due to our oversight, we lost relevant details about the histopathological scoring, for which we sincerely apologize. Considering that this result serves as an auxiliary understanding of the HE staining data and will not significantly affect the core content of this manuscript. After careful discussion, we have deleted this part to ensure the rigor of the manuscript. We have provided detailed responses to other histopathological issues in Comments 5, 6, and 7. 4. As for the TEM result, we will use it for writing another manuscript. 5. We have re-uploaded clearer image of the organizational pathological sections and supplemented its scale. 6. The correlation between morphological observations and the biochemical data, we provided additional information in sections 2.5, 3.1, and the supplementary material.
We sincerely hope this revision will solve the issues you have raised, and we would like to express our heartfelt gratitude once again. Your questions further improved the quality of our manuscript. |
||||||||||||
|
2. Point-by-point response to Comments and Suggestions for Authors |
||||||||||||
|
Comments 1: Line 3. The term hepatopancreas is appropriately used for arthropods and other invertebrates, but not for vertebrates. Although some fish species possess pancreatic tissue embedded within the liver, the hepatic and pancreatic tissues, as well as their functions, remain anatomically and physiologically distinct—unlike the true hepatopancreas of invertebrates, where these functions are integrated within a single organ. |
||||||||||||
|
Response 1: Thank you for your comment. The pancreas of the common carp is diffuse and difficult to separate from the liver, so we generally use the word “hepatopancreas” instead of “liver and pancreas.”There are some relevant references here.We appreciate your feedback and are happy to provide further clarifications if necessary.
1. Zhu, R.;Shang, G.J.;Zhang, B.Y.;Wang, H.T.;Li, L.;Wei, X.F.;Li, D.L.;Yang, Z.Y.;Qu, Z H.;Quan, Y.N.;Liu, S.Y.;Wang, Y.T.;Meng, S.T.;Wu, L.F.;Qin, G.X. Unlocking the Potential of N-Acetylcysteine: Improving Hepatopancreas Inflammation, Antioxidant Capacity and Health in Common Carp (Cyprinus Carpio) Via the Mapk/Nf-Κb/Nrf2 Signalling Pathway,Fish & Shellfish Immunology.2024,144. 2. Li, L.;Wang, Y.T.;Meng, S.T.;Wei, X.F.;Yang, Z.Y.;Zhu, R.;Li, D.L.;Wu, L.F. Effects of Poly-Β-Hydroxybutyrate on Growth, Antioxidant Capacity and Lipid Metabolism of Common Carp (Cyprinus Carpio) under Waterborne Copper Exposure,Aquaculture.2024,580. 3. Li, Z.;Shah, S.W.A.;Zhou, Q.;Yin, X.;Teng, X. The Contributions of MiR-25-3p, Oxidative Stress, and Heat Shock Protein in a Complex Mechanism of Autophagy Caused by Pollutant Cadmium in Common Carp (Cyprinus Carpio L.) Hepatopancreas,Environmental Pollution.2021,287. 4. Wei, X.;Yao, T.;Fall, F.N.;Xue, M.;Liang, X.;Wang, J.;Du, W.;Gu, X. An Integrated Bile Acids Profile Determination by Uhplc-Ms/Ms to Identify the Effect of Bile Acids Supplement in High Plant Protein Diet on Common Carp (Cyprinus Carpio),Foods.2021,10(10). |
||||||||||||
|
Comments 2: Line 29.Do not replicate in keywords, words already present in the title. |
||||||||||||
|
Response 2: Thank you for your reminding. We have turned the keywords into “PBDEs; Transcriptomics and Metabolomics; Multi-omics integrated analysis” in line 30. |
||||||||||||
|
Comments 3: Line 79.Fish biometric data (including length, body mass, age, and, where possible, sex ratio) should be provided, together with the season in which the experiment was conducted, as liver histology is known to vary with season (e.g., hepatic glycogen repletion), sex, and reproductive status. Furthermore, indices such as the hepatosomatic index (HSI) and gonadosomatic index (GSI) could be valuable in detecting gross differences among experimental groups and are generally considered integral components of a well-designed experimental survey involving fish. |
||||||||||||
|
Response 3: Thank you very much for your useful comments.The fish we used were the same batch of juvenile carp, body mass is 22.1 ± 1.3 g. Their reproductive glands had not fully differentiated. So we did not measure the sex ratio and GSI. We are very sorry that the length were not recorded. Thank you very much for your reminder, we will test this data in future related experiments. The season in the experiment was autumn, October to December. As it is difficult to completely classify carp liver and pancreas, we measured the ratio of carp hepatopancreas to body weight. However, this data was used for writing another article, we could provide you with the relevant data, but it is not presented in this manuscript. We have added this information in line 88-90.
We hope this explanation could answer your questions, and we greatly appreciate your valuable comments. |
||||||||||||
|
Comments 4: Line 88. This section should be rewritten for clarity and completeness. The description of the experimental design is currently ambiguous and does not clearly specify the number of fish per group, the number of replicates, or the precise sampling scheme at each time point. From the figures, it appears that there were four experimental groups (control, DMSO, low-dose BDE-47, and high-dose BDE-47), which should be explicitly described in this section. Moreover, the rationale for selecting a 42-day experimental duration should be provided, together with the justification for the chosen BDE-47 concentrations. I consider this a major issue that should be addressed before publication. |
||||||||||||
|
Response 4: We really appreciate your valuable suggestion.We have rewritten this part of the content in line 99-115. “The experiment is divided into 4 groups: blank control group, DMSO group (solvent control group), 40 ng/g BDE-47 group (low concentration group), and 4000 ng/g BDE-47 group (high concentration group), with 3 parallels in each group, totaling 20 carp per parallel, amounting to 360 fish. The carp were randomly assigned to aquariums, with 30 fish in each tank. During the experiment, continuous oxygenation and natural light were maintained, and the carp were fed 3% of their body weight in feed at 14:00 daily. Half an hour after feeding, any uneaten feed and feces were removed. The hepatopancreas, and blood of the carp were collected on day 42 post-feeding, with 12 fish collected from each parallel. After collection, samples were quenched in liquid nitrogen for more than 15 minutes and then frozen at -80 °C for subsequent physiological and biochemical index determination, tissue section preparation, and omics detection. Additionally, to investigate the effect of the feeding duration of BDE-47 on carp, this study also divided healthy carp into 4 groups, with 3 parallels in each group, and 10 fish per parallel. The livers and pancreases of carp were collected on days 1, 3, 7, 14, 28, and 42 post-feeding, with 3 fish collected from each group for subsequent qPCR detection.” The rationale for selecting a 42-day experimental duration: Line 36, PBDEs can exist in the natural environment for several months to several years, still present after 42 days; Line 35, BDE-47 can accumulate in organisms. To observe the chronic toxicity effects of BDE-47 in carp, we chose a long-term experiment. Line 69-71, in our previous study, after 28 days of exposure, some lesions began to appear in the hepatopancreas of the carp , and by 42 days, the lesions were very apparent, so we selected 42 days as the duration. We will supplement this count in our manuscript. The justification for the chosen BDE-47 concentrations: In line 63-64, we added that “Food intake is considered one of the main routes through which vertebrates are exposed to PBDEs.” In line 63, we noted that BDE-47 accumulates more in sediments. In line 66, we mentioned that carp live in the middle and lower layers of water, so they will contact much sediment. Therefore, the concentration of BDE-47 we used refers to the concentrations found in sediment in the aquatic environment and in areas with severe environmental pollution, as mentioned in lines 38 and 39. We hope these instructions can clarify these questions. If needed, we are happy to provide further details. Thank you once again for your feedback. |
||||||||||||
|
Comments 5: Line 103.Which stain? |
||||||||||||
|
Response 5: Thank you for your coment. This stain is Hematoxylin and Eosin staining and Oil Red O staining.We have rewritten this on 2.4. We are grateful for your reminder. |
||||||||||||
|
Comments 6: Line 104. The method should be summarised in this section. In addition, the cited authors (Bernet et al., 1999) themselves refer to other sources, and therefore the relevant methodological details should be explicitly reported here rather than relying solely on secondary citations:
Bernet D, Schmidt H, Meier W, Burkhardt-Holm P, Wahli T. Histopathology in fish: proposal for a protocol to assess aquatic pollution. J Fish Dis. 1999;22:25–34. doi:10.1046/j.1365-2761.1999.00134.x. |
||||||||||||
|
Response 6: Thanks to your very useful advice, we have introduced the methods of H&E and oil red O staining in detail, and supplemented the content in Line 2.4. The hepatopancreas of four-tailed randomly selected carp was removed and fixed with 4% paraformaldehyde for 24 hours. Subsequently, the organs are washed and dehydrated with different concentrations of ethanol. Subsequently, these hepatopancreas are hyalicized in xylene and encapsulated in paraffin. Paraffin blocks were cut into 5 μm slices using a German Leica RM2125 rotary microtome. After dewaxing, the slices were stained with hematoxylin and eosin. Finally, the slices were observed and photographed using an ECLIPSE 200 microscope from Nikon in Japan, and the images were analyzed with Image-Pro Plus 6.0. The scale is 1:200.Three biological replicates were performed in each treatment group, and the number of three tissue sections was analyzed for each biological replicate, for a total of 36 sections, and a total of 6 histological regions were examined. Oil Red O After the liver and pancreas are excised, they are fixed in 10% neutral formalin solution for 10 minutes, washed 2-3 times in PBS, and the surface moisture is blotted dry with absorbent paper. The liver and pancreas from the same location are placed on a fixation tray and quickly frozen in a cryostat. Once the tissue is sufficiently frozen, the liver and pancreas are sliced into thin sections using the cryostat, placed on a glass slide for 10 minutes, and then stained. The sections are immersed in modified Oil Red O staining solution for 15-20 minutes, then washed with isopropanol and distilled water, followed by hematoxylin staining for 5 minutes, and finally differentiated and rinsed. After staining, the slides are mounted with glycerin gelatin for observation and photography. The concentration of IDO in the lipid droplets is measured using Image-Pro Plus 6.0.” Your suggestions have made the experimental methods of our manuscript clearer. Thank you again for your suggestions. |
||||||||||||
|
Comments 7: Line 107-108. Details of the entire procedure should be provided, with particular attention to the number of histological fields examined, the number of tissue sections analysed, the number of specimens (biological replicates) assessed, and the image analysis procedures employed. |
||||||||||||
|
Response 7: Thank you very much for your comment. We have added the relevant content to line 128-133. “The slices were observed and photographed using an ECLIPSE 200 microscope from Nikon in Japan, and the images were analyzed with Image-Pro Plus 6.0. Three biological replicates were performed in each treatment group, and the number of three tissue sections was analyzed for each biological replicate, for a total of 36 sections, and a total of 6 histological regions were examined.” Thank you for pointing out, and I hope our revisions can resolve this issue. |
||||||||||||
|
Comments 8: Line 136. The methods should be summarised here. |
||||||||||||
|
Response 8: Thank you for pointing this out. We have summarized the methods in line 2.10. Thank you again for your valuable suggestion. |
||||||||||||

Round 2
Reviewer 2 Report
Comments and Suggestions for Authors
Unfortunately, the revised manuscript, although objectively improved, still appears as a collection of separate methodological approaches rather than an integrated morpho-functional investigation. Critically, there is no evidence of substantial change in the Results/Discussion section, apart from the improved correlation between lipid droplet accumulation and biochemical data. This remains a major shortcoming, as the lack of integration substantially limits the scientific value and interpretative strength of the study, and it must be addressed before the manuscript can be considered for publication.
As a fish pathologist, I am certainly aware of the widespread but inappropriate use of the term hepatopancreas in fish species such as carp, which indeed harbour pancreatic tissue embedded within the liver. Nevertheless, from a biological and anatomical standpoint, the liver and pancreas remain two distinct organs, in contrast to arthropods, where the true hepatopancreas comprises cells that perform both hepatic and pancreatic functions within a single structure. To clarify by analogy: in carp, the mesonephros harbours three distinct tissues and functions—namely nephrons, haematopoietic tissue, and thyroid follicles—yet it is still correctly referred to simply as the kidney, and not as a “nephrothyroid” or “nephropoietic” organ.
Scientific writing should adopt rigorous and precise terminology. The inappropriate use of terms such as hepatopancreas in fish risks encouraging other authors to replicate the same mistake, thereby propagating conceptual errors in the scientific literature and adding noise to the information base. I therefore strongly recommend that the authors avoid the term hepatopancreas in this manuscript. Instead, they should refer to the organ as the liver, while acknowledging in the text that, in carp, pancreatic tissue is embedded within the hepatic parenchyma.
I appreciate the authors’ effort to clarify the rationale for the 42-day exposure period and the chosen BDE-47 concentrations. However, it should be emphasised that any information directly related to the experimental design—including the justification of the exposure duration and the toxicant concentrations—must be reported in the Materials and Methods section, not in the Introduction. The Introduction should provide context and background, while the Materials and Methods is the appropriate place to describe and justify the experimental design in sufficient detail to ensure reproducibility and proper interpretation. I therefore recommend that the authors transfer this information to the relevant section and present it explicitly alongside the description of the experimental protocol. With regard to BDE-47 administration, the toxicant was delivered orally through incorporation into the feed. However, there is no guarantee that all fish consumed the same quantity of feed and, consequently, the same dose of toxicant. In such cases, it is standard experimental practice to determine the actual concentration of the compound in relevant target tissues (in this case, the liver) and to correlate the biological findings with these analytically measured tissue concentrations. Since the authors did not provide evidence of actual BDE-47 concentrations in liver tissue, this potential source of bias should be explicitly acknowledged and discussed.
Furthermore, the histopathological figures remain inadequate. Despite the authors’ statement that improved images have been provided, the current figures are still of insufficient dimension, magnification, and resolution to permit proper evaluation of the lesions described. Higher-quality images, including representative fields at appropriate magnification, must be provided to substantiate the reported findings. The histological images should be presented separately to ensure adequate size and clarity.
Author Response
Thank you very much for the time and effort you spent on this manuscript. We greatly appreciate the comments you provided, and we have made the following responses to these comments. Hoping that these responses can address the concerns raised. In the attachment, we have provided the manuscript with and without revision marks.Once again, we sincerely thank you for your suggestions.
Thank you very much for pointing out the issues in the manuscript regarding the "integrated morpho-functional". Your comments accurately helped us realize the shortcomings in our research. In our study, we found abnormal metabolite indicators after BDE-47 exposure (Figure 1), further morphological analysis (Figure 2) revealed abnormal cell structures. Cell structure can affect function, combining the results of Figures 1 and 2, we concluded that BDE-47 affects liver cell metabolism and conducted further investigations. We have supplemented relevant content in Lines 132-134 and 146-149. Thank you again for your suggestions.
Thank you very much for your suggestion. We have changed "hepatopancreas" to "liver" in the manuscript, which enhances the rigor of our article. Additionally, we added to line 69-70, "In common carp, pancreatic tissue is embedded within the hepatic parenchyma." Thank you again for your suggestion.
Thank you for your useful suggestions. We have rewritten the 'Materials and Methods' section and added it from Line 418-436. Once again, we appreciate your suggestions.
Thank you very much for your pointed out. Lines 587-594, we explained the lack of rigor regarding the concentration in our experiment. Thank you again for your valuable suggestions.
Thank you very much for your question regarding histopathology. To improve the resolution of the images, we have reorganized them and placed all histopathology-related images separately in Figure 2. In Figure 2, the image stained with HE is at a magnification of 20X, and the image stained with Oil Red O is at a magnification of 40X. The scale bars for both images are presented in the lower left corner. Additionally, we have quantified the results of the Oil Red O staining and provided supplementary information in "3.4". We hope this revision can address your concern, and once again, we sincerely appreciate your question.
Here, we once again express our heartfelt gratitude for the time and effort you have dedicated to reviewing this manuscript. Your meticulous guidance is crucial for enhancing the quality of the manuscript.
Round 3
Reviewer 2 Report
Comments and Suggestions for Authors
I appreciate the authors’ effort to address my previous concerns, and the manuscript has objectively improved. Nevertheless, several issues remain that should be resolved before acceptance.
As a pathologist, I consider the histopathological figures still insufficient to clearly document the reported alterations caused by toxicant exposure. Larger figures are required—preferably occupying at least half a page when presented as a plate. The same recommendation applies to the labelling (arrows, symbols, etc.), which should be of adequate size and clarity to ensure that the lesions are unambiguously recognisable to the reader.
With regard to the authors’ stated explanation concerning the lack of rigour in reporting toxicant concentrations, I was unable to locate this clarification at the cited lines (587–594) in the revised version provided. This discrepancy should be carefully checked and corrected.
Provided that these minor but important issues are definitively addressed, I consider the manuscript suitable for acceptance after minor revision.
Author Response
Thank you very much for your valuable comments on this manuscript, which have improved the quality of our manuscript. We appreciate the time and effort you have dedicated to this manuscript, and we will address the relevant issues raised. We hope that this revision can resolve these problems. In the attachment, we have provided the manuscript with and without revision
marks.
We are very grateful for your professional advice as a pathologist regarding 'histopathologica.' We have adjusted the figures to ensure they are sufficiently large and clear. Additionally, we have labeled the figures with high-contrast colors and provided explanations in the captions. In lines 124-140 (the third paragraph of section 2.1), we have reintroduced the figures. We sincerely thank you for your professional guidance; it was your suggestions that helped us effectively identify and address the shortcomings in the presentation of the images.
We are very grateful for your careful pointing out of this issue, and we sincerely apologize for the inconveniences caused by oversights during our revision process. Regarding the 'rigor of toxin concentration,' we have provided the explanation in lines 592-598 (the second paragraph of 4. Conclusion). Thank you for your corrections that helped us further refine the details of the manuscript.
Once again, we express our heartfelt gratitude! Your feedback has provided important support for enhancing the integrity and credibility of our research. We truly value this feedback and sincerely thank you for your patience and responsibility.